# Single-Stage vs. Multi-Stage Machine Learning Algorithms for Prostate Segmentation in Magnetic Resonance Images

**Author 1**[1]                                                                                   AUTHOR1 EMAIL

**Author 2**[1]                                                                                   AUTHOR2 EMAIL

[1] *address*

## Abstract

Fusion of magnetic resonance images (MRI) with ultrasound has led to major improvements in precision diagnostics for prostate cancer. A key step in the fusion process is segmentation of the prostate in MRI and machine learning (ML) has proven to be a valuable tool for segmentation. In this paper, we compare two ML workflows for prostate segmentation; a single-stage and multi-stage ML workflow to address the challenges of segmentation.

**Keywords:** Machine Learning, Prostate Segmentation, Magnetic Resonance Imaging

## 1. Introduction

One of the recent advancements in the diagnosis of prostate cancer has been the use of multi- parametric magnetic resonance imaging (mpMRI) for identifying targets to biopsy (Bax et al., 2008). Devices such as Artemis$^{\text{TM}}$, in conjunction with ProFuseCAD$^{\text{TM}}$, utilize mpMRI information by co-registering (rigid + elastic) with live ultrasound for a more precise 3D targeted biopsy (Narayanan et al., 2009). In order to co-register between the two modalities, the prostate is segmented in each modality to determine the amount of deformation between the two acquisitions. Numerous methods are available in literature where segmentation is performed automatically or using semi-automated methods (Cuocolo et al., 2019). Recently there has been an increase in using machine learning (ML) to improve robustness and accuracy of segmentation with research showing that ML based segmentation can yield good results in MR segmentation of the prostate (Tian et al., 2018).

A key challenge in the segmentation process is the ability to differentiate the prostate gland from the seminal vesicles. While MR slices through the center of the prostate have clear boundaries, slices near the base of the gland show greater presence of seminal vesicles, leading to poor segmentation. In order to combat this issue, we compare two methods of ML prostate segmentation. The first workflow involves training a segmentation network on a large set of MR data containing images of the prostate with and without seminal vesicles. In the second workflow, the image is classified based on the presence of seminal vesicles and selects an appropriate segmentation network trained on images either containing or not containing seminal vesicles.

## 2. Methods

The models presented utilize two different ML workflows to segment the prostate. The single-stage workflow simply segments the prostate in MR slices regardless of the presence of seminal vesicles, utilizing only one ML model to complete segmentation. The multi-stage workflow classifies the presence of seminal vesicles in an image slice and segments the slice based on the result. The classification algorithm is a convolutional neural network which takes in an image and outputs a binary classification of seminal vesicle presence. Once the determination is made, the image moves to one of two segmentation algorithms which segment the prostate using a U-Net (Ronneberger et al., 2015). Three U-Net segmentation networks were trained with two networks for the multi-stage workflow (with and without seminal vesicles) and one for single-stage (all images).

The training set for the classification and segmentation networks were both acquired from an Artemis database containing 3D MR stacks and physician segmented surfaces. Each prostate slice within the 3D stack was manually labeled as with or without seminal vesicles and the 2D slice was extracted. Then, the corresponding cut through the segmented surface was extracted and converted to a ground truth binary mask. The image extraction and mask creation were performed in MATLAB (Mathworks).

A total of 10,425 image slices were available for training of which 3970 slices had seminal vesicles and 6455 did not have seminal vesicles. For the classification network in the multi-stage model and the lone single-stage network, all images were used for the training and testing of the networks. For the segmentation networks of the multi-stage network, the no seminal vesicle network and seminal vesicle networks were trained and tested on the corresponding data sets. The classification network was trained for 50 epochs with a batch size of 32 images and a learning rate of 0.001995. In each of the three segmentation networks, the images were mean normalized and the networks were trained for 12 epochs with a batch size of 16 and a learning rate of 0.0001. All networks were trained with an Adam optimizer and were developed using Python with the TensorFlow ML library (Google).

## 3. Results

Following the training of the networks, validation was performed on each model and compared against the ground truth labels. To determine accuracy of the classification network, raw accuracy was used, comparing the output of the model to the true label of whether the image slice contained seminal vesicles. On the validation set of 1075 images, the classification network achieved an accuracy of 0.8828.

To assess the performance of the segmentation networks, both mean pixel-wise accuracy and mean Dice similarity coefficient (DSC) were averaged across the test set. For the network trained on images without seminal vesicles in the multi-stage workflow, the model achieved a mean accuracy of 0.9942 and a mean DSC of 0.9105. For the network trained on images with seminal vesicles, the model achieved an accuracy of 0.9917 and a mean DSC of 0.9035. For the single-stage segmentation workflow, the model achieved a mean accuracy of 0.9933 and a DSC of 0.9063.

Figure 1 shows an MR slice overlaid with the single-stage model (blue solid) and multi-stage model (red dashed) contours for an image segmented without seminal vesicles (Figure 1a) and with seminal vesicles (Figure 1b). Contours were extracted from the masks via

MATLAB through edge extraction. Regions of good overlap where the single- and multi-stage workflows match are marked with solid green arrows while regions where the workflows differ are marks with dotted red arrows.

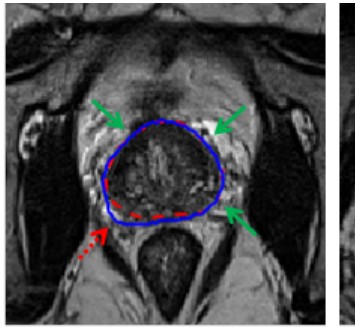 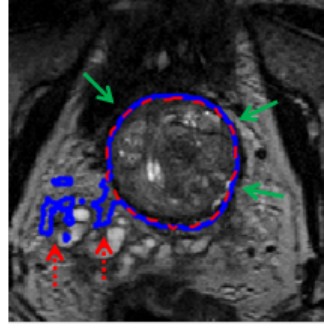

(a) MR Slice without Seminal Vesicles     (b) MR Slice with Seminal Vesicles

Figure 1: a. MR slice without seminal vesicles with single-stage (blue solid) prostate segmentation and multi-stage (red dashed) segmentation from ML model trained on images without seminal vesicles
b. MR slice with seminal vesicles with single-stage and multi-stage segmentation from ML model trained on images with seminal vesicles

On an image without seminal vesicles (Figure 1a), the network trained only on images without seminal vesicles from the multi-stage model aligns with the prostate boundaries slightly better than the single network from the multi-stage model. There are minor discrepancies between the models near the peripheral zone of the prostate with the multi-stage network appearing to better align with the prostate boundary than the single-stage model.

On an image with seminal vesicles (Figure 1), the network trained only on images without seminal vesicles from the multi-stage model aligns with the prostate boundaries much more closely than the single-stage model, particularly posterior to the prostate where the presence of seminal vesicles is most prominent. The network from the single-stage workflow incorrectly selects regions of the seminal vesicles as part of the prostate while the network trained only on images with seminal vesicles from the multi-stage model aligns with the prostate boundaries.

When comparing multi-stage vs. single-stage workflows for prostate segmentation of MR slices containing seminal vesicles, the results presented show there is benefit to a multi-stage workflow. As shown in Figure 1, the networks of the multi-stage workflow are better able to detect the boundaries of the prostate, particularly in regions where prostate tissue is not as differentiated. While this is an extreme case, it does show how the single-stage model can erroneously segment the gland. The classification step should be improved to have a multi-stage workflow which can accurately detecting the presence of seminal vesicles. However, with further improvement to classification and segmentation through hyperparameter tuning or training with stacks of 2D images, the multi-stage workflow could provide further benefit over a single-stage workflow for accurate prostate segmentation.

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
