# OpenReview forum: "Single-Stage vs. Multi-Stage Machine Learning Algorithms for Prostate Segmentation in Magnetic Resonance Images"
_MIDL.io/2020/Conference — Submitted to MIDL 2020_

### Official Review · AnonReviewer3 · 2020-03-10
**lacks of technical novelty and weak experimental results**

**Rating:** 1
**Confidence:** 5

**Review:**

Multi-stage training (with detection network as the first stage) is not new, for example, those with mask RCNN.

Compared with a single-stage approach: with significantly higher costs including additional computation resources (two stages) and manual efforts (labelling each slice w/o seminal vesicles), the overall performance gain seems to be marginal (0.9105, 0.9035 vs  0.9063 in Dice score).

---

### Official Review · AnonReviewer1 · 2020-03-13
**Automatic deep prostate segmentation in MRI**

**Rating:** 1
**Confidence:** 5

**Review:**

This paper presents a pipeline to perform automatic prostate segmentation in MRI. The main assumption is that segmentation performance would benefit from a separate processing of prostate MR images that contain seminal vesicles from those that do not. The proposed architecture consists of a first classification network that is trained to separate images with or without seminal vesicles. Each class of images is then processed through a separate UNet network. This architecture is compared to a standard UNet architecture trained on both types of images.
This paper suffers from several flaws that should be addressed.
Regarding the methodological part:
-The architecture of the classification network should be provided.
-Description of the MRI dataset is critically missing, including  the MRI sequence parameters, scanner, acquisition parameters. A reference to the ‘Artemis’ database should be provided.
-Regarding the evaluation, from what I understand, the authors adopted a resubstitution method (ie train and test on the same dataset) : ”all images were used for the training and testing of the network”. The text should be clarified if I misunderstood. Else, evaluation should be performed in a cross-validation or hold-out scenario to avoid producing optimistically biased results.
 -Regarding the quantitative results, it is not clear if the reported accuracies and DSC for the multi-stage model were estimated from images passed through the segmentation UNet after the classification step or not. If yes, then this means that these values reflect (ie account for) the imperfect accuracy of the classification model (0.8828) which can thus erroneously direct images with vesicles in the no-vesicle UNet and vice versa. If not, then the reported performance only evaluate segmentation performance of each type of images (with or without vesicles). In this latter situation, the authors should perform the whole evaluation accounting for classification step and following a cross-validation strategy as suggested above.
-Quantitative performance by the standard UNet model trained on both types of images are similar to that reported by the two-stage model, thus suggesting that the two-stage model may not be competitive, since it requires to train three deep models instead of one. Please comment.

---

### Official Review · AnonReviewer2 · 2020-03-15
**UNet segmentation of prostate in MRI and ULtrasound for the purpose assisting fusion biopsy**

**Rating:** 2
**Confidence:** 5

**Review:**

VEry small paper. Method not novel. Miss a lot of details to evaluate results.

Single-Stage vs. Multi-Stage Machine Learning Algorithms for Prostate Segmentation in Magnetic Resonance ImagesSingle-Stage vs. Multi-Stage Machine Learning Algorithms for Prostate Segmentation in Magnetic Resonance ImagesSingle-Stage vs. Multi-Stage Machine Learning Algorithms for Prostate Segmentation in Magnetic Resonance ImagesSingle-Stage vs. Multi-Stage Machine Learning Algorithms for Prostate Segmentation in Magnetic Resonance ImagesSingle-Stage vs. Multi-Stage Machine Learning Algorithms for Prostate Segmentation in Magnetic Resonance ImagesSingle-Stage vs. Multi-Stage Machine Learning Algorithms for Prostate Segmentation in Magnetic Resonance Images

---

### Meta-Review · Area_Chair1 · 2020-04-06
**MetaReview of Paper233 by AreaChair1**

**Rating:** 1

**Metareview:**

The reviewers highlighted a lack of novelty and issue on the validation of the performance. Given the doubt on the proper split of the data for training and testing, I recomend this manuscript be rejected.

**Paper Type:**

validation/application paper

---

### Decision · Program_Chairs · 2020-04-11

Reject